# Higher Numbers of Family Meals and Social Eating Behavior Are Associated with Greater Self-Esteem among Adolescents: The EHDLA

**DOI:** 10.3390/nu16081216

**Published:** 2024-04-19

**Authors:** José Francisco López-Gil, Héctor Gutiérrez-Espinoza, David Manzano-Sánchez

**Affiliations:** 1Department of Communication and Education, Universidad Loyola Andalucía, 41704 Seville, Spain; josefranciscolopezgil@gmail.com; 2One Health Research Group, Universidad de Las Américas, Quito 170124, Ecuador; 3Department of Didactics of Musical, Plastic and Corporal Expression, Faculty of Education and Psychology, University of Extremadura, 06006 Badajoz, Spain

**Keywords:** eating habits, education, social identity, family, youth

## Abstract

Background: the aim of this study was to assess the associations of family meals and social eating behavior with self-esteem levels among Spanish adolescents. Methods: This was a secondary cross-sectional study including 706 participants (aged 12 to 17 years; 56.1% girls) from the Eating Habits and Daily Life Activities (EHDLA) study. The evaluation of the frequency of family meals involved participants providing information in physical education classes on how frequently they, along with other members of their household, had shared meals in the previous week. Social eating behavior was assessed by three different statements: “I usually have dinner with others”, “Having at least one meal a day with others (family or friends) is important to me”, and “I enjoy sitting down with family or friends for a meal”. To assess overall self-esteem, the Rosenberg Self-Esteem Scale was used. Results: In the adjusted models, a positive association was observed between the frequency of family meals and the self-esteem score (unstandardized beta coefficient [*B*] = 0.06, 95% confidence interval [CI] 0.003 to 0.12, *p*-adjusted = 0.040). Furthermore, the same positive association was also identified between social eating behavior and the self-esteem score (*B* = 0.23; 95% CI 0.07 to 0.40, *p*-adjusted = 0.005). Conclusions: Although self-esteem is complex and can be influenced by numerous factors, both family meals and social eating behavior may exert a relevant role in adolescents. Encouraging consistent participation in family meals and promoting positive eating practices could be valuable approaches in public health actions targeting the enhancement of self-esteem levels in adolescents.

## 1. Introduction

Self-esteem has been defined as an individual’s subjective judgment of his or her own worth as a person [1]. It is important to note that self-esteem does not necessarily reflect a person’s actual abilities or how others perceive them. Furthermore, self-esteem is often understood as the feeling of being “good enough”, and those with high self-esteem do not necessarily think they are better than others [2]. Boosting self-esteem can be a crucial aspect of prevention and health promotion strategies, as it is a significant risk and protective factor linked to various social and health consequences [3].

The period of adolescence is a critical phase of growth and development marked by significant physical changes and numerous psychosocial transformations, as young individuals seek greater independence from their parents and spend more time with their peers [4]. Adolescents are vulnerable to a range of psychosocial issues, such as mental health problems or low self-esteem [4]. Therefore, self-esteem is widely recognized as a crucial factor in adolescent mental health and development because it reflects an individual’s ability to feel deserving of happiness and effectively cope with life challenges [3]. Moreover, self-esteem plays a crucial role in many of the developmental tasks that adolescents must perform, including the formation of identity [5] and the redefinition of social relationships [6].

Previous research indicates that adolescents with higher self-esteem tend to engage in healthier behaviors such as having good rest habits, consuming less alcohol, maintaining a healthy diet, and respecting mealtimes [7]. Regarding eating habits, they are significantly influenced by the social environment, which includes media, friends, and family [8]. Other people’s presence during an eating occasion or when making food choices can have a considerable impact on behavior [9]. Sharing meals with family and friends is an important daily routine for young people, as it provides an opportunity to connect and strengthen relationships as part of a familiar daily routine [10]. Furthermore, various aspects of the family environment play a crucial role in shaping self-esteem during late childhood and adolescence [11]. For example, family meals, which often reflect family interactions and are influenced by family and societal norms, can promote better self-esteem in young people by fostering indicators of family function, such as increased communication, cohesion, relationship building, and trust [12].

Although a previous systematic review by Harrison et al. [13] revealed a relationship between a greater frequency of family meals and greater self-esteem in young people, to our knowledge, there are no studies assessing the relationship of social eating behavior with self-esteem among adolescents. The increasing knowledge of this association could be relevant since family meals can sometimes occur due to imposition by families rather than due to a predilection for these meals. In fact, dissatisfaction with family relationships or the desire for autonomy are some of the barriers reported by adolescents [14]. In addition, understanding the trajectory of self-esteem throughout life, which typically increases from adolescence to midlife and then decreases in old age [15], highlights the importance of identifying the lifestyle factors that are linked to this psychological dimension in adolescents. Addressing self-esteem in adolescents is crucial, as it constitutes a fundamental aspect of mental health and overall well-being. Examining potential factors related to self-esteem among adolescents (such as family meals, social eating behavior) is not only a public health concern, but also a social issue with far-reaching implications for individual well-being and societal health. Hence, the aim of the present study was to assess the associations of the frequency of family meals and social eating behavior with self-esteem levels among Spanish adolescents.

## 2. Materials and Methods

### 2.1. Sample and Procedure

We utilized data from the Eating Habits and Daily Life Activities (EHDLA) study, which followed the methodology outlined by López-Gil [16]. This secondary cross-sectional study incorporated 706 adolescents (56.1% girls). The EHDLA study included adolescents aged 12 to 17 years from *Valle de Ricote* (Region of Murcia, Spain), and was conducted in all the three secondary schools during the 2021–2022 academic year. All the questionnaires were administered in physical education classes with the presence of one of the study researchers and the physical education teacher.

Before the participants joined the study, their parents or guardians signed an informed consent form, and both the adolescents and their legal guardians received a document outlining the study’s aims, questionnaires, and activities. The adolescents were also required to agree to be involved.

To be eligible for the study, participants needed to be between 12 and 17 years old and either live in or attend school in *Valle de Ricote*. Adolescents who were not enrolled in physical education classes, who had medical conditions that limited their physical activity, or who required medical treatment were excluded because the assessments and questionnaires were conducted during physical education classes. Additionally, adolescents were excluded if parental or guardian consent was not obtained. The study was approved by the Bioethics Committee of the University of Murcia and the Ethics Committee of the Albacete University Hospital Complex (approval IDs 2218/2018 and 2021–85, respectively). This study was conducted following the Helsinki Declaration to safeguard the protection of participants’ rights.

### 2.2. Measurements

#### 2.2.1. Family Meals

Participants were asked about the frequency of family meals in the past week using an ordinal scale ranging from “none” to “seven days” for breakfast, lunch, and dinner. We collected these data separately and summed the reported instances to compute the total frequency of weekly family meals [17].

#### 2.2.2. Social Eating Behavior

We used three statements to measure social eating behavior: “I usually have dinner with others”, “Having at least one meal a day with others (family or friends) is important to me”, and “I enjoy sitting down with family or friends for a meal”. Participants had a 4-point Likert scale with the following response options for each statement: “strongly disagree”, “somewhat disagree”, “somewhat agree”, or “strongly agree”. These responses were scored as follows: “strongly disagree” (1 point), “somewhat disagree” (2 points), “somewhat agree” (3 points), or “strongly agree” (4 points). We summed the responses to these items to calculate a social eating behavior score, which ranged from 3 to 12 points. A greater score indicated a higher level of social eating behavior. In the EAT (Eating and Activity over Time) Project [18], the reliability of these statements was acceptable (i.e., Cronbach’s alpha [*α*] = 0.70). 

#### 2.2.3. Self-Esteem

Rosenberg’s Self-Esteem Scale was used to evaluate overall self-esteem [2]. This scale consists of 10 items rated on a 4-point Likert scale (ranging from 1 to 4), with a total score ranging from 10 to 40 points. Higher scores indicate greater self-esteem. The scale’s reliability has been validated, particularly among adolescents [2]. The translated and validated Spanish version of the scale, which has demonstrated satisfactory internal consistency and temporal stability, was used [19].

#### 2.2.4. Covariates

The study participants were requested to provide their sex and age. The Family Affluence Scale (FAS-III) [20] was utilized to determine their socioeconomic status (SES), with scores ranging from 0 to 13 points. The adolescents’ body weight was measured on an electronic scale, with an accuracy of 0.1 kg (Tanita BC-545, Tokyo, Japan), with participants wearing minimal clothing. A portable stadiometer was used to measure height, with an accuracy of 0.1 cm (Leicester Tanita HR 001, Tokyo, Japan). Their body mass index (BMI) was calculated by dividing the weight in kilograms by the height in meters squared. Furthermore, BMI z-scores were computed according to the World Health Organization criteria [21]. The Spanish version of the Youth Activity Profile (YAP-S) [22], a 15-item self-report questionnaire, was used to assess physical activity and sedentary behavior. This tool, validated for Spanish youth [23], categorizes activities into school, outside school, and sedentary habits, using a 5-point Likert scale to reflect the previous week’s activities. To assess sleep duration, participants were asked to provide their usual bedtime and wake-up times on weekdays and weekends. The average daily sleep duration was calculated using the following formula: [(weekday sleep duration × 5) + (weekend sleep duration × 2)]/7. A self-administered food frequency questionnaire (FFQ), validated for the Spanish population, was utilized to estimate energy intake. 

### 2.3. Statistical Analysis

To evaluate the normal distribution of the variables, we utilized visual methods like density and quantile–quantile plots, along with conducting the Shapiro–Wilk test. Therefore, continuous variables are shown as a median and interquartile range (IQR), and categorical variables are shown as counts and percentages. Given the absence of specific cutoff points of family meals and social eating behaviors, these variables were converted into tertiles. Therefore, for family meals status, the following groups were established: low family meals (0 to 11 meals), medium family meals (12 to 15 meals), or high family meals (16 to 21 meals). For social eating behavior status, the next categories were determined: low social eating behavior (0 to 9 points), medium social eating behavior (10 to 11 points), or high social eating behavior (12 points). Linear regression models were conducted to calculate the unstandardized beta coefficient (*B*) and its 95% confidence interval (CI) for the associations of the family meal frequency and social eating behavior scale (both as continuous and categorical variables) with self-esteem. These models were fitted using robust methods because they offer several advantages in the context of dealing with heteroscedasticity and outliers [24]. The method “SMDM” was applied, which resulted in an initial S-estimate, followed by an M-estimate, a Design Adaptive Scale estimate, and a final M-step. Moreover, we calculated the estimated marginal means of the self-esteem score according to the number of family meals or the points on the Social Eating Behavior Scale, as well as for their categories. The models were adjusted for age, sex, SES, physical activity, sedentary behavior, overall sleep duration, body mass index, and energy intake. We carried out all the analyses with R statistical software (version 4.3.3) (R Core Team, Vienna, Austria) and RStudio (version 2023.12.1+402) (Posit, Boston, MA, USA). A *p* value of less than 0.05 was selected as a threshold for statistical significance.

## 3. Results

The descriptive data of the adolescents included are presented in Table 1. The median weekly family meals and social eating behavior score of the adolescents were 14.0 (IQR = 6.0) and 10.0 (IQR = 2.0), respectively. In addition, the overall mean self-esteem score was 26.0 points (IQR = 5.0) on a scale ranging from 10 to 40 points.

The unadjusted and adjusted unstandardized beta coefficients of the association between the frequency of family meals and social eating behavior among adolescents are shown in Table 2. Considering family meals and social eating behavior as continuous variables, a positive association was observed between the frequency of family meals and the self-esteem score (*B* = 0.071; 95% CI 0.008 to 0.133, *p* = 0.026), as well as for social eating behavior (*B* = 0.259; 95% CI 0.091 to 0.427, *p* = 0.003). However, although these associations were positive, they were all low. On the other hand, when examining family meals and social eating behavior as categorical variables, compared to adolescents with a low number of family meals, a positive and significant association was identified among those with a high number of family meals (*B* = 1.100; 95% CI 0.365 to 1.835, *p* = 0.004) (after adjusting for several covariates). Furthermore, the adjusted analysis for social eating behavior showed that, in comparison with adolescents with a low level of social eating behavior, a positive and significant association was observed in their counterparts with a high level of social eating behavior (*B* = 1.190; 95% CI 0.390 to 1.990, *p* = 0.004). 

The estimated means (for the unadjusted and adjusted models) for the self-esteem score for each further family meal or point on the social eating behavior scale are reported in Figure 1. In the adjusted model, the means self-esteem score ranged from 26.0 (95% CI 24.8 to 27.2) for those with the lowest score on the Social Eating Behavior Scale (i.e., three points) to 28.1 points (95% CI 27.6 to 28.7) for those with the highest score on this scale (i.e., 12 points). Similarly, in terms of the frequency of family meals, the adjusted model showed that the means self-esteem score ranged from 26.8 (95% CI 25.8 to 27.7) for adolescents without family meals to 28.2 points (95% CI 27.5 to 28.9) for their counterparts with the highest number of weekly family meals (i.e., 21 family meals).

Figure 2 displays the estimated means (for the unadjusted and adjusted models) for the self-esteem score according to the family meal or social eating behavior status. Regarding family meals, the adjusted model identified that the highest self-esteem score was in adolescents with high numbers of family meals (*M* = 28.4; 95% CI 27.6 to 28.9), while the lowest was in those with low numbers of family meals (*M* = 27.1; 95% CI 26.5 to 27.8). Furthermore, significant differences were identified between adolescents with high family meals and those with low (*p* = 0.004) or medium family meals (*p* = 0.033). In relation to social eating behavior, the adjusted model reported the highest self-esteem score in those with high levels of social eating behavior (*M* = 28.5; 95% CI 27.7 to 29.2). Conversely, the lowest self-esteem score was found in adolescents with low levels of social eating behavior (*M* = 27.3; 95% CI 26.8 to 27.8). Significant differences were observed between adolescents with high social eating behavior and those with low (*p* = 0.004) or medium social eating behavior (*p* = 0.043).

## 4. Discussion

Overall, our research indicates that both the frequency of family meals and social eating behavior are related to higher self-esteem in adolescents. Notwithstanding, the effect sizes observed for these associations were low, suggesting that other lifestyle-related (e.g., drug use, social networks use) or social factors (e.g., family connectedness, neighborhood) could also account for higher levels of self-esteem in adolescents. Our findings are in line with previous scientific studies [13,25,26], indicating that sharing meals can foster a supportive network that enhances positive self-esteem. For instance, Utter et al. [26] identified that parents with zero to two family meals per week had a lower self-esteem level (based on the Rosenberg’s Self-Esteem Scale which ranged from 6 to 24 points) (*M* = 18.8; 95% CI 17.9 to 19.8), than their counterparts with three to six family meals per week (*M* = 19.2; 95% CI 18.4, 20.0), and those who had family meals every day (i.e., seven times per week) (*M* = 20.4; 95% 19.6 to 21.1). Similarly, Eckert et al. [25] found that, in comparison with children who ate family supper five or more times per week, children who ate family supper never or less than once per week had greater odds of having low self-esteem (OR = 1.97; 95% CI 1.51 to 2.56). In addition, Fulkerson et al. [27] observed that the Family Connectedness Scale (*B* = 0.16; *p* < 0.010) and Positive Meal Atmosphere Scale (*B* = 0.19; *p* < 0.001) (among other family variables) were associated with higher self-esteem among adolescent girls with obesity. Another study among adolescents indicated that those reporting eating five to seven family dinners per week were more likely of having high self-esteem (OR = 1.40; 95% CI 1.27 to 1.49) compared to those who reported eating zero to one family dinners per week (according to the Profiles of Student Life: Attitudes and Behaviors Survey) [28]. Conversely, Eisenberg et al. [29] reported lower odds of having low self-esteem per each further family meal in both adolescent boys and girls (boys: OR = 0.96, 95% CI 0.87 to 1.06); girls: OR = 0.96, 95% CI 0.90 to 1.04). Notwithstanding, these associations were not significant after adjusting for several covariates (e.g., family connectedness, race/ethnicity, socioeconomic status). It is important to exercise caution when comparing our findings to those of this study due to the different tools used to assess self-esteem (i.e., Rosenberg’s Self-Esteem Scale or Profiles of Student Life: Attitudes and Behaviors Survey) and the diverse ways of analyzing both family meals and self-esteem (i.e., continuously, with arbitrary cut-off points, etc.). Supporting this notion, it must be considered that most of these studies only examined the relationship between the family meal frequency and self-esteem with no other aspects of family meals, such as social eating behavior, which may contribute to the protective effect. This limitation hinders our understanding of the unique mechanisms that contribute to the protective effects of family meals [12]. Although family meals are associated with health and well-being benefits [13], further research is needed to fully explore the causal relationship [30]. This approach is also applicable for social eating behavior [31]. Despite this lack of causal evidence in the scientific literature, there are several possible reasons for these results.

First, the role of emotional and social support during family meals or social eating could be one possible explanation for these findings. Engaging in family meals and practicing social eating behaviors can foster an environment of emotional and social support for youths [32], which may contribute to higher self-esteem levels [33]. Although self-esteem also influences the amount of support parents provide, the supportive behavior of parents affects children’s self-esteem [34]. In support of this idea, parental support, emotional warmth, and positive parent–child relationships have been found to be significant factors related to a child’s self-esteem [35]. Moreover, in adolescents, family meals may also enhance health and well-being (including self-esteem) [29].

Second, family meals and social eating experiences (such as those with family or friends) can serve as a platform for open communication and interaction among adolescents [36]. Parents who have conversations about everyday things can establish a fundamental sense of ease, promoting overall communication, which could create a comfortable foundation for addressing more delicate and challenging issues [12,37]. The communication that occurs during shared meals can contribute to building healthy relationships that enable adolescents to express their thoughts, feelings, and concerns [38]. Feeling heard and understood can boost adolescents’ self-esteem by addressing various aspects of their complex and multifaceted self-concept [39]. Additionally, mealtime routines involve practical communication about tasks that need to be completed, serve as a brief time commitment with minimal conscious thought, and are repeated regularly over time [36]. Furthermore, research suggests that family meals can provide a buffer against daily risks associated with familial conflicts, leading to greater happiness and role fulfillment in adolescents [40]. All these factors may (at least partially) account for the higher levels of self-esteem observed in adolescents.

Third, another possible reason for the observed results may be related to the strengthening of social identity. Family meals and social eating interactions can significantly shape the identity of adolescents [41]. Sharing food possesses an aggregative capacity that reinforces the social connections and shared identity among individuals partaking in a meal [42]. Studies indicate that family meals offer an opportunity for demonstrating behaviors or roles to others and the social learning of eating habits and behaviors, contributing to the preservation of familial bonds and collective responsibilities within the family [30]. Furthermore, having a sense of belonging and strong connections with family and friends can enhance self-esteem by offering a sense of safety and approval [11]. According to social identity theory, adolescents with higher levels of social eating behavior may experience enhanced self-esteem by affiliating with groups, taking pride in group achievements, and fostering a positive group image [43], thereby reinforcing their self-image and overall self-esteem [44]. On the other hand, adhering to a group’s standard can be a gratifying experience [45], and sharing a meal with others might enhance the pleasurable aspects of the activity [46]. It is possible that adolescents exhibiting greater levels of social behavior derive increased satisfaction from these rewarding experiences, reinforcing their motivation and consequently boosting their self-esteem [47].

Fourth, engaging in shared meals during adolescence can foster the development of social skills [48], which can lead to higher levels of self-esteem [49,50]. Family meals can be determinants for establishing and enriching social relationships [26,28]. Mealtime conversations offer an opportunity for parents to socialize their children and teach valuable communication skills and manners [51]. Research has shown that social skills training can improve self-esteem among adolescents [49,50]. Additionally, social eating behaviors can foster the development of social skills and help individuals understand how social structural properties impact food choice practices [48]. Positive relationships between the family meal frequency and internal assets, such as social resistance skills, having a sense of purpose, and a positive view of the future, suggest that adolescents may learn social skills and develop more positive self-esteem during mealtime interactions [28].

This study has several limitations that should be taken into account. For instance, the cross-sectional design impedes to establish cause and effect relationships. Therefore, longitudinal studies are required to determine whether more frequent family meals and higher levels of social eating behavior directly contribute to a rise in self-esteem in adolescents. Additionally, the experimental designs could elucidate the mechanisms underlying these relationships. Furthermore, relying on self-reported data opens the door to potential recall and social desirability biases, which can adversely affect the accuracy of the frequency of family meals, social eating behavior, or self-esteem. Moreover, some cultural factors were not controlled for and could affect the associations found. Investigating the role of cultural factors in shaping mealtime practices and their potential impact on self-esteem is crucial for developing culturally sensitive interventions. On the other hand, this study has several strengths. First, to our knowledge, there are no studies assessing the association between social eating behavior and self-esteem among adolescents. Second, this study contributes cross-sectional evidence concerning the role of family meals and social eating behavior in self-esteem among an understudied population (i.e., adolescents). Third, we adjusted our analyses for various covariates (i.e., sociodemographic, anthropometric, and lifestyle variables), which enhances the reliability of the findings. Nevertheless, even with these adjustments, it is crucial to acknowledge that there may still be residual confounding that cannot be totally eliminated. Moreover, self-esteem is multifactorial, and although our analyses are adjusted for these covariates, it is possible that other factors not considered in this study may influence the results observed.

## 5. Conclusions

Although self-esteem is complex and can be influenced by numerous factors, both family meals and social eating behavior may exert a relevant role in adolescents. Given the growing body of evidence highlighting the important real-world impact of self-esteem [52] and the role that shared meals appear to play in its development [13], it seems necessary to acknowledge the potential influence of these social and familial factors. Encouraging consistent participation in family meals and promoting positive experiences of eating practices could be valuable approaches in public health actions targeting the enhancement of self-esteem levels in adolescents. Furthermore, integrating food education into school curriculums and providing psychosocial support for adolescents experiencing difficulties with self-esteem may offer additional possibilities for intervention.

## Figures and Tables

**Figure 1 nutrients-16-01216-f001:**
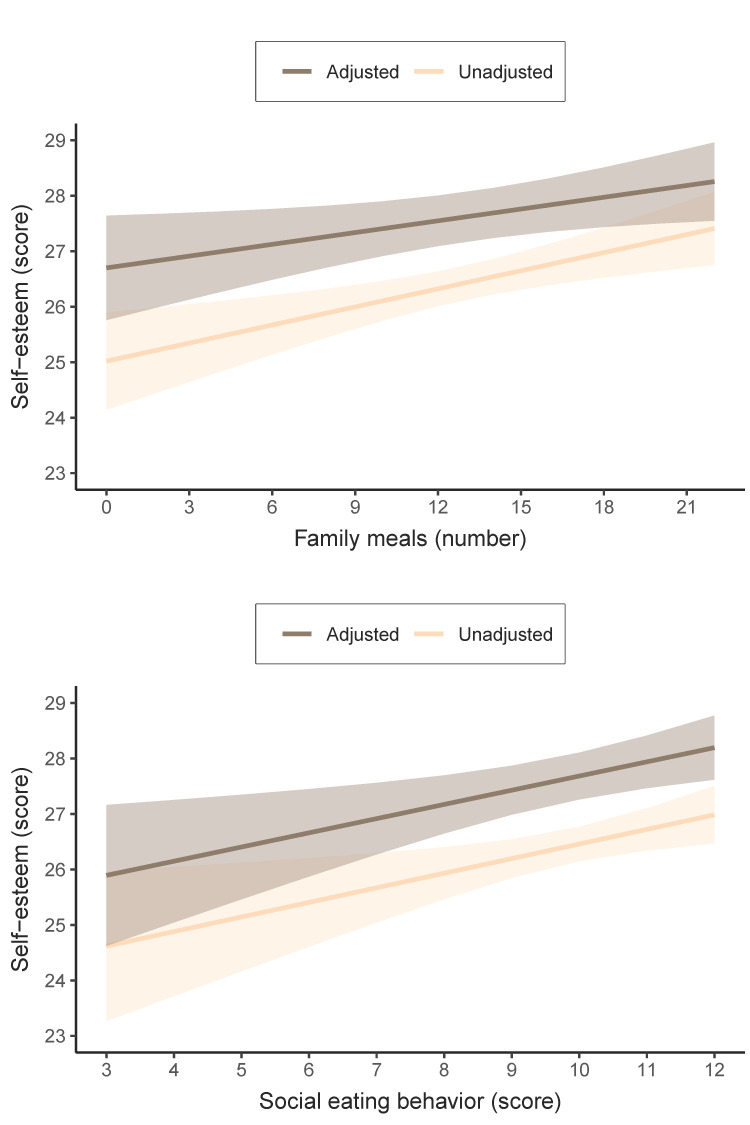
Estimated means of self-esteem score for each further family meal or point to the social eating behavior scale in adolescents. Adjusted for age, sex, socioeconomic status, physical activity, sedentary behavior, overall sleep duration, body mass index, and energy intake.

**Figure 2 nutrients-16-01216-f002:**
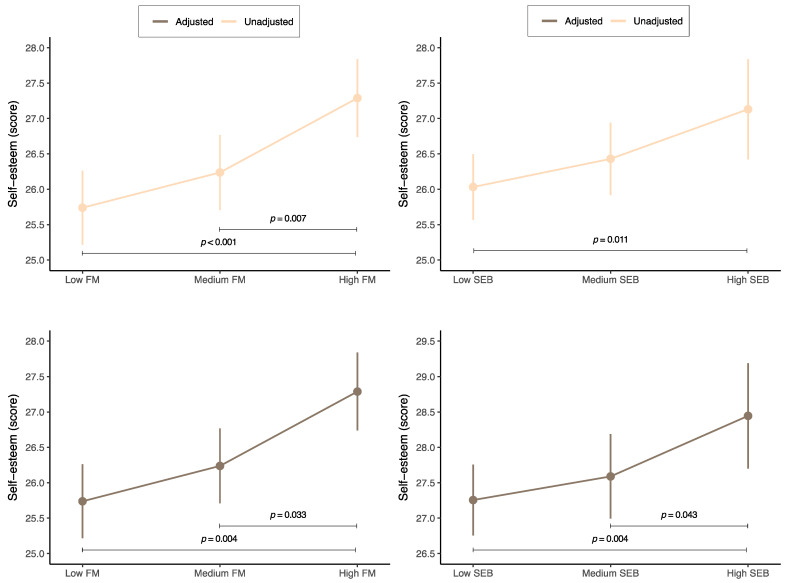
Estimated means of self-esteem score based on family meal or social eating behavior status among adolescents. FM, family meals; SEB, social eating behavior. Adjusted for age, sex, socioeconomic status, physical activity, sedentary behavior, overall sleep duration, body mass index, and energy intake.

**Table 1 nutrients-16-01216-t001:** Main characteristics of the study participants (*N* = 706).

Variables		Total *
Age (years)	Median (IQR)	14.0 (2.0)
Sex	Boys (%)	310 (43.9)
	Girls (%)	396 (56.1)
FAS-III (score)	Median (IQR)	8.0 (3.0)
YAP-S physical activity (score)	Median (IQR)	2.6 (0.9)
YAP-S sedentary behavior (score)	Median (IQR)	2.6 (0.8)
Overall sleep duration (minutes)	Median (IQR)	501.4 (71.8)
Body mass index (kg/m^2^)	Median (IQR)	21.7 (6.0)
Body mass index (z-score) ^†^	Median (IQR)	0.0 (2.0)
Energy intake (kcal)	Median (IQR)	2554.3 (1465.9)
Weekly family meals (number)	Median (IQR)	14.0 (6.0)
Social eating behavior (score) ^‡^	Median (IQR)	10.0 (2.0)
Self-esteem (score) ^§^	Median (IQR)	26.0 (5.0)

* Data are expressed as the median (interquartile range) or count (percentages). FAS-III, Family Affluence Scale-III; IQR, interquartile range; YAP-S, Spanish Youth Active Profile. ^†^ According to the World Health Organization criteria [21]. ^‡^ Social eating behavior scale ranges from 3 to 12 points. ^§^ According to Rosenberg’s Self-Esteem Scale (10 to 40 points) [2].

**Table 2 nutrients-16-01216-t002:** Unadjusted and adjusted unstandardized beta coefficients of the relationship of family meal frequency and eating social behavior with self-esteem in adolescents.

	Self-Esteem (Score) ^†^	
Predictor (continuous)	*B* (95% CI, *p* value) (unadjusted)	*B* (95% CI, *p* value) (adjusted ^‡^)
Family meals (per further meal)	0.109 (0.045 to 0.173, *p* = 0.001)	0.071 (0.008 to 0.133, *p* = 0.026)
Social eating behavior (per further point)	0.270 (0.095 to 0.445, *p* = 0.003)	0.259 (0.091 to 0.427, *p* = 0.003)
Predictor (categorical)	*B* (95% CI, *p* value) (unadjusted)	*B* (95% CI, *p* value) (adjusted ^‡^)
Low number of family meals (0–11 weekly meals)	Reference	Reference
Medium number of family meals (12–15 weekly meals)	0.499 (–0.244 to 1.242, *p* = 0.188)	0.295 (–0.415 to 1.005, *p* = 0.415)
High number of family meals (16–21 weekly meals)	1.550 (0.791 to 2.309, *p* < 0.001)	1.100 (0.365 to 1.835, *p* = 0.004)
Low level of social eating behavior (3–8 points)	Reference	Reference
Medium level of social eating behavior (9–11 points)	0.404 (–0.294 to 1.102, *p* = 0.258)	0.336 (–0.328 to 1.000, *p* = 0.323)
High level of social eating behavior (12 points)	1.120 (0.285 to 1.955, *p* = 0.009)	1.190 (0.390 to 1.990, *p* = 0.004)

^†^ According to Rosenberg’s Self-Esteem Scale (10 to 40 points) [2]. ^‡^ Adjusted for age, sex, socioeconomic status, body mass index, physical activity, sedentary behavior, overall sleep duration, and energy intake. *B*, unstandardized beta coefficient; CI, confidence interval. Models were fitted using robust methods (i.e., “SMDM” method) [24].

## Data Availability

Data generated or analyzed during this study are available from the corresponding author upon reasonable request, since they pertain to minors, and privacy and confidentiality must be respected.

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
