# Peer review of "Higher Numbers of Family Meals and Social Eating Behavior Are Associated with Greater Self-Esteem among Adolescents: The EHDLA"

_nutrients, 2024, doi:10.3390/nu16081216_

Round 1
Reviewer 1 Report
Comments and Suggestions for Authors
Dear authors,
Because the topic is quite interesting an useful, I suggest you to improve a bit the introduction section by offering some statistics regarding the problems related to food, if there are any. More, I suggest to improve the discussion by highlighting the achievements within the highlight of previous researches. You should explain better the implications of the study and the future directions
Author Response
Reviewer 1
Dear authors,
Because the topic is quite interesting an useful, I suggest you to improve a bit the introduction section by offering some statistics regarding the problems related to food, if there are any. More, I suggest to improve the discussion by highlighting the achievements within the highlight of previous researches. You should explain better the implications of the study and the future directions.
Thank you for your comment. We have included further information in the discussion section regarding previous studies on this topic. Furthermore, we have updated the implications of this study and future directions.
Reviewer 2 Report
Comments and Suggestions for Authors
This is a well written manuscript but can be improved.
Introduction: This is a well written introduction, however, there needs to be more emphasis on why it is important to examine self esteem and social eating behavior particularly among adolescent age group. It is not clear whether this is a public health issue or a social issue that warrants attention.
Method: This section needs additional statistical analysis. The Rosenberg scale is a likert scale response, then why was a ordinal regression or multinomial regression not performed. The IQRs do not specify the association between the two selected variables. How was social eating frequency assessed as an exposure variable is not clear, responses such as 'sometimes' 'usually' are quite vague and are open to different interpretations. Instead, having meals three times a week with family is a more absolute measurement of the variable.
When adjusting for socio demographic variables, why was the 'Income' variable not considered, because it can have a significant impact on having family meals. Neighborhood may also play a role in this association.
Results: This section is too small and should be elaborated with results from additional analysis.
Author Response
Reviewer 2
This is a well written manuscript but can be improved.
Thank you for your time and feedback.
Introduction: This is a well written introduction, however, there needs to be more emphasis on why it is important to examine self esteem and social eating behavior particularly among adolescent age group. It is not clear whether this is a public health issue or a social issue that warrants attention.
Thank you for your indication. The next information has been added: “Addressing self-esteem in adolescents is crucial, as it constitutes a fundamental aspect of mental health and overall well-being. Examining potential factors related to self-esteem among adolescents (such as family meals, social eating behavior) is not only a public health concern but also a social issue with far-reaching implications for individual well-being and societal health”.
Method: This section needs additional statistical analysis. The Rosenberg scale is a likert scale response, then why was a ordinal regression or multinomial regression not performed. The IQRs do not specify the association between the two selected variables. How was social eating frequency assessed as an exposure variable is not clear, responses such as 'sometimes' 'usually' are quite vague and are open to different interpretations. Instead, having meals three times a week with family is a more absolute measurement of the variable.
Thank you for your comment. Although the scale includes Likert scale items, responses are commonly computed to obtain an overall score, where higher scores indicate higher self-esteem. For this reason, we consider it appropriate to treat this variable as continuous.
Furthermore, we reported the IQR to describe our continuous variables and understand the dispersion of variables, not to observe the association between variables. For that purpose, we use generalized linear models.
On the other hand, for a better understanding, we have included further information about “social eating behavior”.
Lastly, we have included further analyses using our main predictors (family meals and social eating behavior) as categorical variables. Given that previous studies have reported different arbitrary cut-off points (0-2, 3-6, and 7; 5-7 or less than 5-7, etc.). Therefore, we have converted these variables into tertiles.
When adjusting for socio demographic variables, why was the 'Income' variable not considered, because it can have a significant impact on having family meals. Neighborhood may also play a role in this association.
Thank you for your indication. As socioeconomic status indicator, we used the Family Affluence Scale-III which was designed in the context of the Health Behaviour in School-aged Children Study (HBSC), a project sponsored by the World Health Organization (WHO) and involving almost 50 Western countries. Furthermore, we do not have information about the neighborhood (unfortunately), so we have specified this limitation.
Results: This section is too small and should be elaborated with results from additional analysis.
Thank you for your comment. We have added further analyses (Table 2 and Figure 2) and more information in the results section.
Round 2
Reviewer 1 Report
Comments and Suggestions for Authors
Dear authors,
from my point of view your paper is improved and ready to be published